# Visual Cultural Biases in Food Classification

**DOI:** 10.3390/foods9060823

**Published:** 2020-06-23

**Authors:** Qing Zhang, David Elsweiler, Christoph Trattner

**Affiliations:** 1Institute for Language, Literature and Culture, University of Regensburg, Universitätsstrße 31, 93053 Regensburg, Germany; david.elsweiler@ur.de; 2Department of Information Science & Media Studies, University of Bergen, Fosswinckelsgt. 6, 5007 Bergen, Norway; christoph.trattner@uib.no

**Keywords:** visual biases, food classification, crowdsourcing

## Abstract

This article investigates how visual biases influence the choices made by people and machines in the context of online food. To this end the paper investigates three research questions and shows (i) to what extent machines are able to classify images, (ii) how this compares to human performance on the same task and (iii) which factors are involved in the decision making of both humans and machines. The research reveals that algorithms significantly outperform human labellers on this task with a range of biases being present in the decision-making process. The results are important as they have a range of implications for research, such as recommender technology and crowdsourcing, as is discussed in the article.

## 1. Introduction

Visual processing plays a significant role in human decision making [1] but can be biased in several ways. For example, limited cognitive capacity means we are inclined to focus on the most salient elements of stimuli and filter out other aspects [2]. This, in turn, means that the presentation of visual cues can bias the decisions people make. Good examples of this are signs in supermarket shelves that improve the salience of products and increase their sales as a result [3], or the placement of items on a restaurant menu that make certain meals more likely to be chosen [4]. Visual biases of this type transfer to digital environments. Chen and Pu, for instance, discovered that patterns of visual attention change according to the layout of a recommender interface [5]. In our study, we focus on cultural differences in visual biases related to food. The reasons for focusing on food are twofold: first, food is central to human health and quality of life and thus the problems most related to our work, food identification and food recommendation, are both problems that have received significant research attention in recent years. Second, past research has shown that in food identification tasks, algorithms can outperform human users [6]. The reasons why this is the case, however, are not particularly well understood. We postulate that human biases, such as those described above, may be playing a role. To our knowledge very little research has been performed in this area as most work has focused on dataset biases and how these may be resolved, e.g., [7,8]. There has been some prior work, however, that has explored how known human visual biases, such as canonical perspective (the preference for seeing objects from a certain perspective) [9] and the Gestalt laws of grouping (the tendency to see objects as collections of parts) [10] can be used to improve object classification [11]. Our work is different because we examine how human biases harm classification accuracy and not improve it, focusing on the classification of food images.

A second component of our work is the attempt to understand the cultural influence on how visual biases impact human decisions. Again, limited literature exists on this aspect. Vondrick et al. [11] did show the existence of cultural differences in visual biases. In their work they demonstrated that people from different cultures had varying mental visual representations of objects, which could be harnessed to improve classification performance. Again, our work is different because we examine this kind of bias in detail, focusing on the classification of foods sourced from different food cultures. It is well-known that food preferences vary geographically, both across [12] and within countries [13]. This also applies to visual preferences for food [14], with scholars arguing that if such cultural-related context factors are ignored when developing recommendation systems, biased (and therefore poorer) recommendations will be provided [14]. This makes the relationship between the origin of the food and the individual to whom it should be recommended an important one. It is within this context that we study participants’ perception of recipes.

In this article, we present a study whereby participants from three countries, China, USA, and Germany, are asked to label images of food. The labels they apply are the country from which they believe the recipe was sourced. Studying a task with a known “true label”, and collecting predictions from both algorithms and human judges, we can achieve the following objectives:Determine how able humans are to categorise recipes by origin.Understand the visual and other factors which influence (and bias) the labels they apply.Compare the performance of humans and machine learning algorithms for this task.

In line with our objectives this work aims to answer three research questions:*RQ1*. To what extent is it possible to classify recipes from the recipe portals of different food cultures with machine learning models based only on visual properties?*RQ2*. How able are humans to distinguish recipes from the recipe portals of different food cultures solely by observing the recipe images?*RQ3*. Which factors (i.e., information cues from the images or user properties) influence the judgements made?

## 2. Materials and Methods

### 2.1. Data Collections

The recipes and associated images studied were sourced from three popular recipe portals from China, Germany and the US. We collected 25,508 recipe images from *Xiachufang.com*, 35,501 from *Allrecipes.com* and 72,899 images from *Kochbar.de*. Recipes from *Xiachufang.com* were crawled from the website during the period 22–26 October 2018, whereas the images and recipes from *Allrecipes.com* and *Kochbar.de* were re-used from our past work [15]. These are amongst the most popular recipe sharing websites in China, the US and Germany, respectively. In all cases we stored only one image for each recipe, taking the initial, default associated image. To ensure equal classes we randomly selected 25,000 images from each portal for our analyses.

### 2.2. Food Classification by Means of Visual Features and Machine Learning

To establish the extent to which it is possible to use visual information to automatically determine the portal from which a recipe was sourced, we formulated the problem as a prediction task whereby classifiers were trained to predict the source portal for each image. The images were represented as a multi-dimensional vector by extracting 5144 visual features from each image. The idea was to generate as many features as possible that may capture elements of what participants perceive and utilise when assigning labels. The features, described in detail below, include explicit visual features (EVF), colour histogram, local binary patterns (LBP), descriptors from the scale-invariant feature transform algorithm (SIFT), and deep neural network image embeddings (DNN).

#### 2.2.1. Explicit Visual Features (EVF)

The first set of features, which we refer to as explicit visual features (EVF), were originally proposed by San Pedro and Siersdorfer [16]. The ten features in this set represent low-level image properties including image brightness, sharpness, contrast, colorfulness, entropy, RGB (Red Green Blue) contrast, variation in sharpness, saturation, variation in saturation and naturalness. These features are simple to calculate and have shown utility in several image popularity predictions and recommendation tasks, from the photos in Folksonomies [16] to specific categories of images, such as recipe images [15] and artwork [17]. The freely available OpenIMAJ (http://openimaj.org) framework was employed to calculate the EVF features.

#### 2.2.2. Colour Histogram

Colour can strongly influence human perception of food and alter eating behaviours [18]. Colour has even been shown to affect human judgements with respect to the other sensory properties of food, such as taste or flavour [19]. To capture the colour properties of an image, images can be represented as colour histograms, which describe the global distribution of colour in the image. We computed a multi-dimensional colour histogram in the RGB colour space, which simultaneously represented three colour channels with eight bins per colour channel. This resulted in an 8 × 8 × 8 = 512-dimension vector for each image. This form of representation has shown utility in both image classification (e.g., [20]), and retrieval tasks (e.g., [21]).

#### 2.2.3. Local Binary Patterns (LBP)

LBP describes images in their entirety by computing the local representation of texture. Proposed by Ojala et al. [22], LBP has been employed in several domains including facial recognition [23], image retrieval [24] and object detection and matching [25] owing to its ability to discriminate and isolate changes. LBP ignores colour information. Before extracting, therefore, original images are transformed into grey scale. Pixels from the image are then selected randomly and the grey value of 24 neighbours in a circle with the radius 8 pixels around these are compared. If the grey value of the chosen pixel is greater than or equal to one of its neighbours, the neighbour point is set to 1. Otherwise, the point gets a value 0. Subsequently, a group of binary strings are formed, and the LBP value of the chosen pixel is the decimal converted from it. The process is repeated until the LBP value has been computed for every pixel. The final features describing the texture of the image are obtained by counting the frequency of LBP values. Here, we employed uniform LBP, which is defined as the LBP with only at most 2 transitions from 0 to 1 or vice versa; others were deemed to be one situation. Since 24 neighbours for each pixel are chosen, a vector of 24 + 2 = 26 dimensions was calculated.

#### 2.2.4. Descriptors of Scale-Invariant Feature Transform (SIFT)

SIFT is a further robust local image representation [26]. The main idea of using SIFT is to identify and describe the keypoints within images. Keypoints represent a sparse set of image regions that contain a complex image gradient structure. Following the approach described in [27] to identify these, we applied to each keypoint a 128-dimension descriptor. Since each image had a different number of keypoints, however, the dimensions of the visual features of each image were not of equal size. As such, we applied k-means clustering (k = 500) on all descriptors, and the centre of each cluster was deemed a codeword and could be used to form a codebook. The final step was calculating the frequency histogram of each codeword in the codebook for each image; those frequency histograms formed the bag of visual words (BoVW), inspired by the bag of words model in Natural Language Processing [28]. In the end, each image was represented by a 500-dimension vector.

#### 2.2.5. Deep Neural Network Image Embeddings (DNN)

Deep learning has widely applied in diverse fields with promising results. In terms of image classification, several deep neural networks have been developed, such as AlexNet [29], GoogLeNet [30], ResNet [31], etc., which have proven to be powerful in a number of tasks, from medical applications, such as identifying cancerous cells [32], to urban planning [33]. In the food domain, such models have been used to improve accuracy in food categorisation [34] and to estimate the nutritional content of a meal [35]. Inspired by these developments, we applied VGG-16, which is a deep neutral network pretrained with ImageNet [36], which has shown impressive predictive power in food image retrieval [6]. We extracted the features of layer fc1 from VGG-16 by using the Keras (http://keras.io) framework, resulting in a 4096-dimensional vector for each image.

After extracting the visual features, each image in our dataset was transformed into a 5144-dimensional vector and represented by the feature sets described above. We built classifiers by using each feature set individually and then all feature sets as a combination. Three supervised classification approaches were applied: naive Bayes (NB), logistic regression (LOG) and random forest (RF). In all experiments the data were split randomly into training (70%) and testing (30%) sets, with a fivefold cross-validated randomised search cross-validation being applied on the training set to select the optimal parameters for logistic regression and random forests.

### 2.3. Food Classification by Means of Human Judgement

To establish human performance on the same task we designed a remotely deployed experiment and recruited participants via crowd-sourcing platforms and social media. The experiment was hosted on a server owned by the University of Regensburg, Germany and in all cases accessed by means of an anonymised URL. By recruiting participants located in China, the United States and Germany, this allowed us to study the influence of culturally induced biases.

#### 2.3.1. Study Design

In the main part of the study participants were shown images sourced from different portals and were required to answer 3 questions with respect to each image. On completing the study, participants provided demographic and other background information. Participants were each shown 9 images, 3 from each dataset, one after the other. All images were drawn randomly from the same test set used to evaluate our classifiers (see above). To increase the generalisability of the findings, we maximised the number of images used by assigning each image to only one participant. After showing an image, participants were first asked to decide from which of the three recipe portals the associated recipe was sourced. The study approach, the selection of the images, the questions asked, and their wording were tested in a small-scale pilot study prior to performing these experiments.

Next, participants were asked to report, on a 5-point Likert scale, their confidence in the label they assigned. In a final question, participants were able to select one or more items from a list of factors that we believed may have been influential in their judgements. These included factors relating to food, e.g., recognisable ingredients, type of food, food colour and shape, as well as non-food factors, such as the food container, eating utensils or their gut instinct. The reasons for focusing on these factors are that they are commonly reported in the literature and reflect features of our classification approaches. More concretely: *Ingredients:* The ingredients of meals are commonly used to build food classifiers, e.g., [37,38].*Type:* As shown in [39], when food type is given, it is helpful for algorithms in predicting food ingredients. We put the factor Type here to see if food type has a positive influence for the human in making the judgement.*Colour:* Colour is also often used to classify food automatically [40] and in our case corresponded to the visual feature of colour histogram. The colour of food has also been proven to affect human perception of food, sometimes leading to misrecognition [18,41].*Shape:* This relates to the visual feature LBP. According to [42], humans rely on shape in classifying objects while algorithms pay more attention to texture.

While the above listed factors all relate to the food itself, the remaining questions were associated with supplementary factors, such as container, eating utensils and instinct, which were all reported by the participants as important during the pilot survey.

Participants could also list further factors in a free-text field. An example task and associated questions are shown in Figure 1.

After labelling the images, participants completed the study by answering 13 questions, which captured participant demographics as well as other information of interest. The details are shown in Table 1.

#### 2.3.2. Participants

The study was originally deployed on Amazon Mechanical Turk (MTurk: https://www.mturk.com/), a popular crowdsourcing platform, as a means to recruit participants restricted to individuals from China, the US and Germany. To ensure participants performed reliably, participation was restricted to only those who had a “HIT accept rate” of more than 98% in their previous tasks. Participants were paid USD 0.50 for their participation. This approach quickly provided the sought-after 100 participants from the US, but after several weeks only 57 German participants were recruited, and no Chinese participants were found. To recruit German participants, we supplemented our sample by advertising via university mailing lists (our institution is located in Germany) and social media via the authors’ personal Twitter (https://www.twitter.com/) and Facebook (https://www.facebook.com/) accounts. We additionally deployed a Chinese version of the study (where instructions and questions were translated to Chinese) on the platform Wenjuanxing (https://www.wjx.cn/) and advertised this on the Chinese social media channels Douban (https://www.douban.com/), Xiaomuchong (http://www.xiaomuchong.com/bbs/) and Wechat. Participants were reimbursed RMB 1 for taking part. These approaches combined allowed us to recruit 100 participants from each country. In the end, 300 participants from the three countries were recruited. Figure 2 shows the distribution of the participants’ age (Figure 2a) and gender (Figure 2b) from each location. Participants who were located in Germany and China were younger than those in the US and the distribution of gender in each country was also imbalanced. More males took part in the US and Germany, while this trend is reversed in the Chinese sample.

#### 2.3.3. Methods of Data Analysis

After the collection phase was complete, the data were analysed in different ways. The classification performance of both the prediction models and human judgements was measured in terms of accuracy (ratio of successfully made classifications to total number of classification decisions, ACC). The performance of both the prediction models and human judgements was visualised using confusion matrices. These are useful since they help illustrate in which cases mistakes were made, as well as how these were made (i.e., which labels were erroneously applied in which cases). Appropriate inferential statistics were used to establish differences across groups (e.g., in terms of gender, interest in food/recipes from foreign cultures, etc.). Binary logistic regression analyses were applied to determine if participants’ answers related to demographic or other factors and ordinal logistic regression models were built with the same factors, as well as participants’ reported confidence in their labels, to understand which factors help predict confident decisions. Binary logistic regression was used in cases where the dependent variable had two classes; ordinal logistic regression was employed when the dependent variable was measured on an ordinal scale. We created numerous different models using groups of feature sets as shown in the tables in appropriate sections below.

Participant responses to free-text questions were analysed qualitatively using a bottom-up, inductive approach. Responses were coded in duplicate, similar or related responses were grouped together, and the groups were collapsed until a hierarchical structure was formed. We communicate the results in the form of a coding scheme and provide examples to illustrate the most important codes.

## 3. Results

The results of our experiments are reported in the following subsections to answer the three questions we raised in Section 1.

### 3.1. Classifying the Origin of Recipes Based on Visual Properties with Machine Learning Approaches (RQ1)

Table 2 presents the performance of each classifier. The bottom line of the table illustrates that the recipe images from the three recipe portals are sufficiently visually distinct, such that they can be classified by the algorithms with relatively high accuracy. When using all of the visual features available, all three classifiers offered accuracy (ACC) of ACC = 0.73 or better with the logistic regression model achieving the highest accuracy of ACC = 0.89. The DNN features offered the best predictive power while SIFT was ranked in second place. Single EVF features offered the lowest accuracy, but nevertheless, all performed slightly better than random (ACC = 0.33). Models utilising all EVF features offered improved accuracy (ACC = 0.47–0.55). The performance of the remaining feature sets such as colour histogram and LBP shows no significant difference when combining EVF.

Figure 3 shows the confusion matrix for the best performing model, illustrating that the classifier was more accurate when identifying recipes from *Xiachufang* (ACC = 0.95) than classifying those from the other two (ACC = 0.86 and 0.85). The majority of misclassifications for *Allrecipes* and *Kochbar* were labelled as belonging to the other of these two classes, with very few being misclassified as *Xiachufang* recipes. In other words, when applying the same algorithms and visual features to images, the recipes from the Chinese recipe portals seem easier to differentiate.

In summary, the experiments show that it is possible to distinguish between the recipes from different recipe portals of China, US, and Germany based solely on the proposed visual features. *Xiachufang* recipe images appear to be more visually distinct with images from the other two portals more likely to be confused.

### 3.2. Analysing Human Labelling Performance (RQ2)

As shown in Figure 4, human performance on the same food classification task was markedly poorer. Figure 4a presents the accuracy distribution over all 300 participants, with most achieving an accuracy of between ACC = 0.40 and 0.60; M = 0.49. Figure 4b depicts how accuracy varied for participants from the three countries across the different food portals. Performance for the Chinese and American participants was highest when they were tasked with classifying recipe images from their own country. Participants from China were particularly accurate with *Xiachufang* recipe images, with the accuracy ACC = 0.67. Participants from Germany, on the other hand, achieved a slightly higher accuracy when classifying recipes from *Xiachufang* than images from *Kochbar*, with the ACC = 0.55 and 0.54 respectively. For Chinese and German participants, recipes from *Allrecipes* were the most difficult to classify.

When comparing the performance of our human participants to those achieved by the algorithms above (i.e., by examining the confusion matrices in Figure 3 and Figure 5), we see that humans make choices biased in the same direction as those generated algorithmically. Figure 5, which provides the confusion matrix of their judgements, indicates that participants made more mistakes when classifying recipes from *Allrecipes* and *Kochbar*. More than 30% of recipes from *Allrecipes* were identified as from *Kochbar*, while 10% fewer were mistaken for recipes from *Xiachufang*. Participants behaved similarly when classifying the recipes from *Kochbar*. At the same time, more than half of the recipes from *Xiachufang* were classified correctly. The human judgements, therefore, followed the same trend as those provided by the algorithms: the images from *Xiachufang* seemed to be most visually distinct, whereas those from *Allrecipes* and *Kochbar* seemed to be most similar.

Participants from different locations displayed diverse degrees of confidence in each recipe portal, as shown in Figure 6a. In general, participants reported higher confidence when labelling recipes sourced from the country where they reside. This is particularly true for the participants from USA and Germany. Moreover, both the German and US participants reported least confidence when labelling images from *Xiachufang*. The findings may shed light on cultural differences with respect to confidence, with the Chinese exhibiting caution rather than confidence and the participants from the United States exhibiting high confidence in their judgements other than for images from the Chinese site.

Figure 6b presents the correlation matrix for the confidence scores participants applied to their labels for images sourced from different recipe portals. It demonstrates that participants’ confidence in their labels for *Allrecipes* and *Kochbar* images correlated positively (*p* < 0.05), while a negative correlation existed between the confidence in labels for both western portals and *Xiachufang* images. This finding aligns with those described above. It seems that when participants assumed a recipe originated from *Xiachufang*, they then believed that it was unlikely to come from the other two recipe portals and vice versa. In other words, participants believed recipe images on the western portals to look similar to each other, but different to those from *Xiachufang*.

To summarise, in this section we have shown that participant performance in the labelling task was significantly poorer than the machine learning approaches in the previous section. The analyses, moreover, reveal differences in the labels applied and the performance of participants from different countries for images sourced from different portals. Participants typically performed best and were more confident when labelling images sourced from their home country.

### 3.3. Factors Leading to or Influencing Participants’ Judgements (RQ3)

In this section we explore the labelling decisions made by participants in detail. We do this by first looking at the visual features, which proved useful when predicting the source of an image, to determine if the same information can help predict the labels applied by participants. Next, we examine the explanations participants gave for their choices to understand how choices were made and/or biased, as well as to determine which, if any, helped lead to a correct label being applied. Lastly, we examine how labelling performance varied across different groups, which provides an insight into how demographic variables can influence the way images of food are perceived.

#### 3.3.1. Predicting Participant Label Based on Visual Features

Table 3 presents the utility of various visual components with respect to (a) predicting a recipe’s origin and (b) predicting the label applied to the image by participants in the experiment. The first thing we notice when examining Table 3 is that visual information features tell us more about the actual source of a recipe image than the label applied to it by the participant. The highest accuracy for image source achieved was ACC = 0.84 with a combined feature set, which was slightly lower than with the full test set (see Section 3.1) achieved when attempting to predict participant judgements. The best performance achieved an accuracy of ACC = 0.46, again using all of the visual features available. This is an initial indication that participants were not using the same visual properties as the algorithms to make their decisions.

#### 3.3.2. Participant Explanations for Labelling Choices

The lower part of Table 3 demonstrates how classifiers performed using the predefined explanations we provided to participants to justify their performance as features. As can be read from the table, none of these features were helpful, either for predicting origin or the labels participants assigned. Most likely this was because the explanations did not advocate for a specific class, e.g., some utensils (for example, chopsticks) may have indicated Chinese food, whereas others may have been a sign of a western dish.

Table 4 shows the frequency with which the most common factors and combination of factors were selected by participants to justify the labels they applied. The ingredients featured in the image, type of food and the combination of these two features were the most commonly reported as influencing decisions. These findings underline that although participants were only presented with visual information in the form of an image, the labelling choice was made based on a semantic interpretation of the image content. Moreover, in 127 cases participants reported making decisions based on “instinct”, that is, a feeling that the recipe was sourced from a particular recipe platform. Colour and shape—the two obvious visual properties listed—seem to have been supplementary factors, since, as shown in Table 4 and Figure 7, they were more likely to be chosen with other factors rather than being chosen alone. Factors such as container and eating utensil were selected least frequently, although it is important to note that not every image contained a container or utensil.

#### 3.3.3. Free-Text Explanations

Participants were also able to provide additional descriptions to justify their decisions in their own words using free-text comments. A total of 14 participants from China, 33 from the US and 22 from Germany provided 166 such explanations, which were analysed qualitatively in a bottom-up fashion as described above. Duplicate, similar or related responses were grouped together, and the groups were collapsed until a hierarchical structure was formed. The coding scheme for the factor is shown in Table 5.

Two high-level categories were discovered: food-based and non-food-based. Non-food factors included watermarks, commonly used date formats for specific countries, or objects or background aspects surrounding the pictured meals, which helped the participants make judgements.

Both food and non-food factors featured aesthetic dimensions, which may be related to the visual aspects represented in the machine learning features. Comments categorised as Adjective, Style or Photo were somehow related to visual aspects. Several participants described the recipe images aesthetically and treated photography as the basis for judgements, e.g., “Angle of the photo, light in the photo” (US_72). On the other hand, other justifications required abstraction or reflection on the images to derive semantic properties, including what ingredients a meal contained, how it was cooked, how it may taste, whether or not it was healthy, etc. Some participants even reported how their personal experiences with this kind of food influenced the label they assigned. All of these factors underline how the participants’ knowledge and background influenced or biased the label they applied.

The free-text comment box was occasionally used by participants to explain their uncertainty. We assigned these cases most often to the category “Text”. We examined the images in these cases manually and discovered that they all originated either from *Xiachufang* (see Figure 8a) or *Kochbar* (see Figure 8b). Most of the texts were added with post-processing, as shown in Figure 8a: the uploaders tagged the recipes with the dish names or their usernames. Similarly, the brands on the food packages revealed information related to recipes’ origins, like the images on the left of Figure 8b: these brands are common in German supermarkets but rare in the other two countries. Texts offer concrete information for humans, and as such the accuracy of participants in such cases increased to ACC = 0.94.

#### 3.3.4. Factors Leading to Correct Classification Choices

To determine which factors aided participants in classifying recipes correctly, we developed further logistic regression models. To do so, cases where labels were assigned correctly were given a value of 1 and cases where an incorrect label was given, 0. This value was then used as the dependent variable in the analysis. The predictors (independent variables) were the predefined explanatory factors described above. The results are shown in Table 6.

Only food type and eating utensils proved to have a significant (*p* < 0.05) influence on participants’ ability to label images correctly. We must acknowledge, however, that the fit of the model is not particularly strong, as indicated by the low R^2^ value. That being said, when participants reported noticing eating utensils, prediction accuracy increased from ACC = 0.48 to ACC = 0.57. The increase was especially pronounced for recipes from *Xiahucfang*, where accuracy increased from ACC = 0.53 to ACC = 0.75. To exemplify why performance increased in such cases, recipes with eating utensils originating from *Xiachufang* are shown in Figure 9. These were all classified correctly by our participants; the traditional Chinese eating utensils chopsticks are obvious in the images, which increased the probability of participants labelling correctly.

In a next step, we investigated whether the same factors had an influence on participants’ confidence that they were labelling images correctly. For this, ordinal regression models were used, one model per collection, the results of which are shown in Table 7.

The first thing to observe is that different features were found to be helpful for different collections. Type, container, eating utensils and instinct were useful predictors for confidence when *Xiachufang* was to be judged; for *Allrecipes*, colour, eating utensils and instinct were significant features; and for *Kochbar* only the presence of ingredients was found to be a significant feature.

The only features with positive coefficients, i.e., features that when present increased participant confidence, were found in the model for *Xiachufang*. When a participant reported the presence of a container or eating utensil, on average this increased their confidence in the label applied. The remaining significant features were indicators which reduced confidence. In other words, acknowledging the presence of certain ingredients in a recipe from *Kochbar* tended to lower confidence in the assigned label on average. We also note that while the presence of eating utensils increased confidence for *Xiachufang* recipes, the trend was the opposite for images from both the other collections. Moreover, when participants reported making a decision based on instinct in all three collections this resulted in lower confidence ratings on average, which makes sense.

#### 3.3.5. Varying Performance across Participant Groups

To understand if participant demographic information influenced their ability to determine the portal from which a recipe originated, we examine how the accuracy of participants’ judgements varied on each recipe portal depending on how they answered the post-experiment questionnaire. Table 8 presents the results, revealing that participants with different ages and genders behaved differently when judging recipes’ origins. Younger participants (<35) achieved higher accuracy when labelling recipes from *Xichufang* (ACC = 0.59 vs. ACC = 0.49) but they performed significantly worse than older participants in labelling *Allrecipes* (ACC = 0.41 vs. ACC = 0.52).

Female participants achieved higher accuracy on *Xiachufang* (ACC = 0.61 vs. ACC = 0.51) while they underperformed compared to male participants on *Kochbar* (ACC = 0.44 vs. ACC = 0.51). We must interpret the findings regarding age cautiously, though. As the sample age distribution in our samples varies across countries, it is very possible that the effects found relating to age are simply a consequence of participants being best able to identify foods sourced from the portal in their home country.

An additional question invited the participants to share their travel experiences and experiences of each country. This allows us to understand whether the classification decisions participants made varied according to their experience of being in the other countries. Analysing the data revealed that accuracy did not increase as a result of frequent cross-continental travel. People who had lived in a country for a longer time were, however, significantly better able to classify the recipes from the portal of that country. Other observations include that participants who had spent time in China were more accurate when labelling recipes from *Allrecipes*, whereas those with more experience of the US were less accurate when labelling *Xiachufang* images. Less surprisingly, being familiar with the recipe portal influenced the accuracy of judgements. Participants who reported being more familiar with *Allrecipes* provided significantly more accurate judgements on recipes from this portal. Familiarity with *Xiachufang* and *Kochbar*, on the other hand, had no significant influence on the accuracy of images from these portals. Participants unfamiliar with *Allrecipes* and *Kochbar* were better at judging the recipes from *Xiachufang*.

Participants who reported being interested in food or recipes from foreign cultures achieved higher accuracy overall. Similarly, those participants who reported trying food from other cultures were also more accurate in the labelling task.

The analyses in this section have shown that it was not only the participants’ culture that influences the labels that they applied. Individual traits and personal experience also played a role in the labels that were assigned, and the accuracy achieved.

## 4. Discussion and Conclusions

The analyses reported in the previous section, shed light on how visual-based choices can be influenced by diverse factors including cultural differences, but also by a range of other contextual properties. We focused on the task of labelling foods with a particular location because of the importance of food to human life and the visual nature of food choices.

In the first step, we compared the performance of human judges from the countries with automated classifiers employing machine learning approaches. Next, to better understand how the participants interpreted the image visual cues they were presented with, we attempted to use the same machine learning approaches to understand which features helped predict the labels participants assigned. Finally, we examined the performance of participants from different groups with different demographics and properties across images from the three collections. The results of the analyses performed help answer our research questions, introduced in Section 1. We summarise the insights gained in relation to the research questions below:

In response to *RQ1* our experiments show that classification algorithms can achieve high accuracy when determining the source of recipes based solely on visual properties of the image associated with a recipe. Almost all of the image properties tested provided some useful signal for this task, the strongest being provided by DNN. Overall, the images from the Chinese recipe portal were labelled most accurately, with recipe images from The US and German portals more likely to be confused. The results show that the Chinese-sourced images were more visually distinct than those from *Allrecipes* and *Kochbar*.

Our results show that humans are far less accurate at the same task. While in the literature there is evidence that for other food classification tasks the best performing algorithms can perform comparably with human labellers [6], our findings, for this particular task, are even stronger. The evidence suggests that unlike the machine learning approaches, humans abstract or interpret the visual features to derive semantic features, such as the ingredients a meal contains or how it may taste. As this process is based on personal knowledge or experience the act of classification becomes biased, which evidently negatively influences accuracy. When humans made classification errors, however, the trend in their mistakes was the same as for the machine learning approach. The Chinese-sourced images were more likely to be accurately labelled, while those from the German and US sites were more likely to be confused. The confidence levels associated with the labels applied confirm that the participants were aware of this trend. It is not easy to compare our findings to past results from the literature given the specific nature of the tasks studied. The task studied in our case—determining the source of a recipe—is much more challenging than that studied in [6], which made it ripe for identifying the biases involved. Moreover, unlike in [11], the visual biases we uncovered did not improve human classification performance, but rather hindered it.

Underlining the diverse biases at play in the labelling task, the experiments showed that predicting the labels participants applied turned out to be a much more challenging machine learning task than predicting the actual source website for the recipe. The performance of human labellers was substantially poorer than the algorithms. The collected data shed some light as to why this was the case. The participants reported several features of the images as being influential when making their decisions although some justifications were more useful than others. The features dominant in the literature for food perception tasks, such as colour [18,41] and shape [42], were less important than the ingredients present and type of dish. Our results show that if the participants recognised the dish type from the image, it was more likely that they made the right choice. Moreover, participants were able to improve their performance by identifying factors in the image which had nothing to do with the food itself, but offered discriminative power. Eating utensils, such as cutlery or chopsticks, or text being present in the image were prominent examples. The results, moreover, demonstrate that participants with different demographics perform differently on this task. Experiences of the relevant culture and familiarity with the recipe portal both had an influence on participant accuracy. The modelling work identified other demographic factors that superficially appear to be important, such as age and gender. We posit, however, that differing sampling mixes across the countries mean that these are largely tied to interest in and experience of the relevant food culture.

### 4.1. Implications of the Results

In this section we discuss what we believe to be the implications of our results. We relate our findings to the problem of food recommendation, which is our main area of interest, but we also raise a note of caution with respect to the use of crowd-sourcing platforms when collecting data for food identification tasks.

Our findings underline that the way people perceive images of food differs fundamentally based on different factors. The primary factor we studied was the participant’s country of residence and we discovered that this directly influenced the labels applied to images in the study. While we did not study food preference directly, our findings do have consequences for the development of food recommendation systems since familiarity with food—and visual familiarity in particular—is strongly related to food preference [43,44]. The foods people find desirable—and to what extent they are willing to try something new—are tightly bound to their cultural upbringing and to physical and emotional reactions to food experiences in the past [43], but also depend on individual traits, such as openness to experience [45]. We also note in our findings that the perception of images and the resulting labels were correlated with several demographic factors, such as familiarity with the recipe portals and interests in food and recipes from foreign cultures.

This reinforces the need for food recommendation systems to model and account for contextual variables when making personalised food recommendations. Our results also offer an explanation as to why—in contrast to many other domains, such as music or film recommendation—standard recommendation technologies do not perform well for the recommendation of food [46].

Certainly, more research is required to understand which contextual factors are important and how these can best be modelled and incorporated in recommendation algorithms. Our findings underline the importance of culture as a dimension in combination with other demographic factors. Initial work in this direction has been initiated in the domain of music recommendation (e.g., [47]), but no equivalent research exists for the recommendation of food.

The results here additionally have implications for the collection of data for food identification research using crowdsourcing platforms, such as Amazon Mechanical Turk. Crowdsourcing has become popular in diverse research areas because it can be used to recruit a large sample of workers in a short period of time for relatively little financial outlay. This method was used in the largest dataset available for food identification [48]. However, as our results show, caution is necessary when taking this approach. Differing cultural backgrounds, personal experiences and interests will influence how food images are perceived. Moreover, as our experience with recruiting through Amazon Mechanical Turk showed, it is challenging to ensure diversity in participants. This problem has been noted by other scholars who are working to address this issue algorithmically [49].

### 4.2. Limitations of the Study

There are several limitations to our work that we wish to acknowledge. To maximise the number of images tested, and thus the generalisability of our findings, our experiments were designed such that images were only labelled by a single participant. This has the disadvantage that we have no means to compare labels applied across participants or groups of participants. In future work we aim to complement the analyses here with a design that allows multiple judgements for single images to be compared as in [50,51].

A second limitation to note is the presence of text in some of the images, which, as reported above, influenced the labels assigned by some participants. Based on the free-text explanations provided by participants, text appeared only in the images sourced from *Xiachufang* and *Kochbar*, with 30 and 19 recipe images with text being reported in these portals, respectively. Although we reported the use of this text as a finding, it was not our intention to study such images.

Building on this work, our future research will explore whether similar cross-cultural biases are present when users apply subjective labels to recipes. We plan to employ a similar experimental setup but collect data on participants’ subjective impression of recipes (e.g., their attractiveness, how willing they are to cook and eat them, etc.). This would complement the findings presented in this paper nicely and would offer concrete utility with respect to the design of food recommendation systems.

In this work we have explored the influence of contextual factors on the way people perceive images of food. In our experiments, where human annotators and machine learning algorithms labelled images of food, the algorithmic approach outperformed the human labellers by a large margin. Further analyses reveal several reasons why annotators misclassified, including basing judgements on factors that are coloured by past experience and knowledge.

## Figures and Tables

**Figure 1 foods-09-00823-f001:**
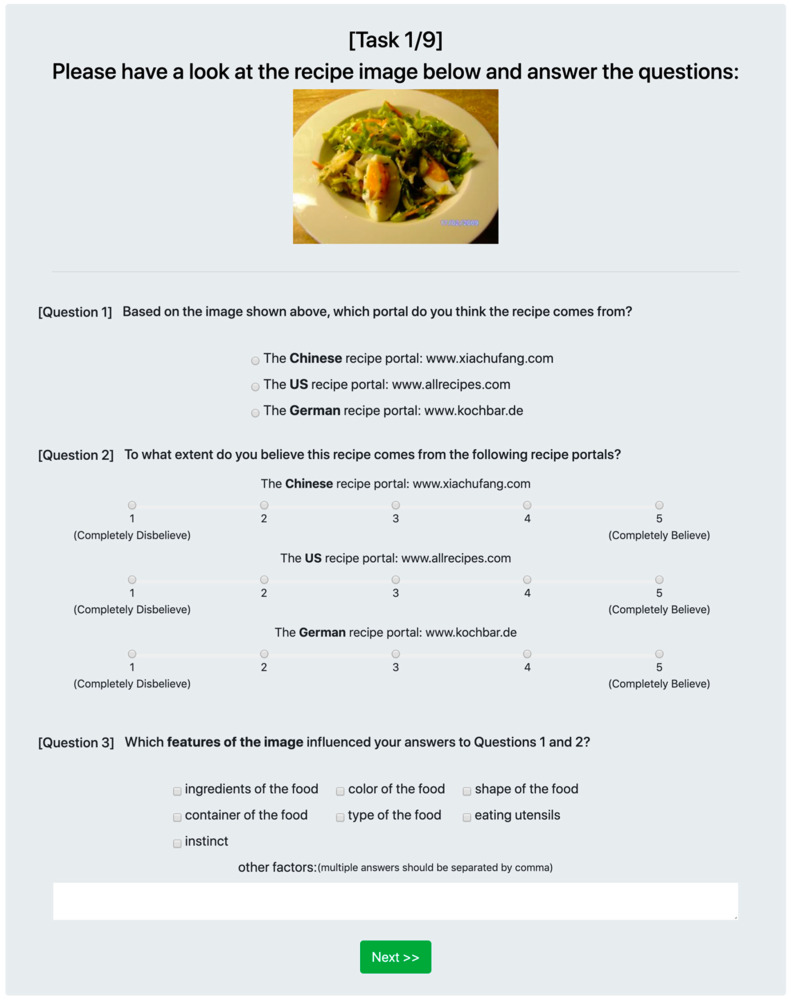
Example of the online survey.

**Figure 2 foods-09-00823-f002:**
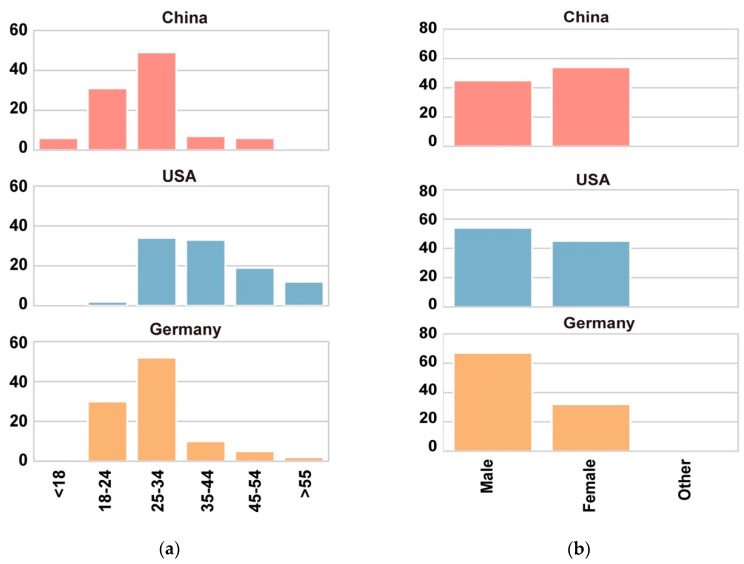
Study participants’ demographics. (**a**) Age distribution of participants from each country. (**b**) Gender distribution of participants from each country.

**Figure 3 foods-09-00823-f003:**
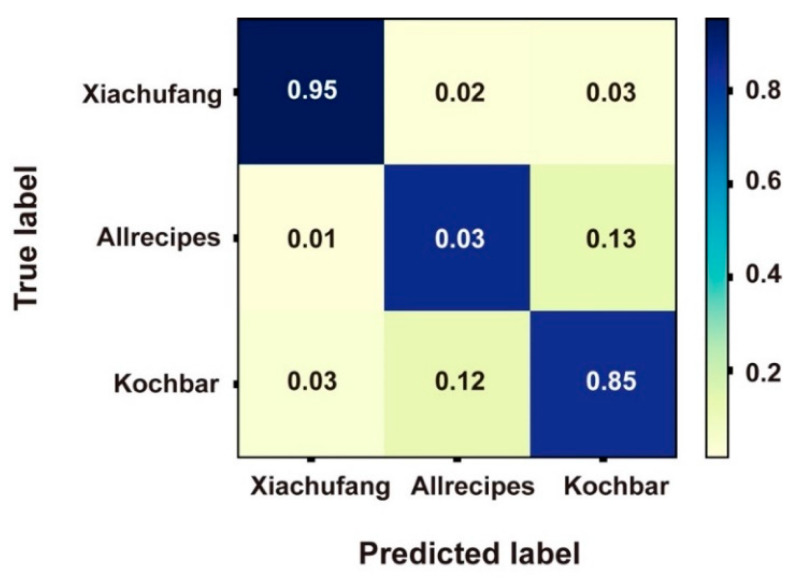
Confusion matrix of the best performing classifier on the samples.

**Figure 4 foods-09-00823-f004:**
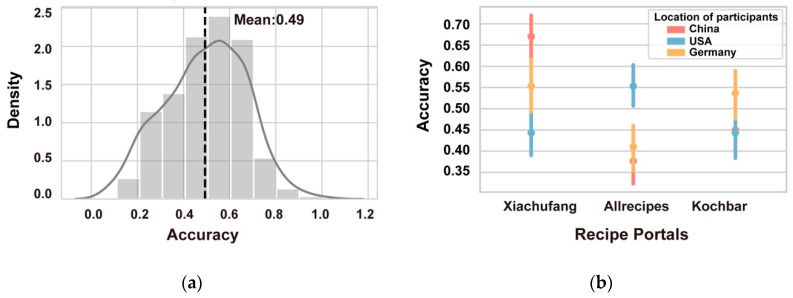
Human performance on food origin classification task. (**a**) Distribution and mean value of participant accuracy. (**b**) Mean value and error bar for participant accuracy for each recipe portal, grouped by participant origin.

**Figure 5 foods-09-00823-f005:**
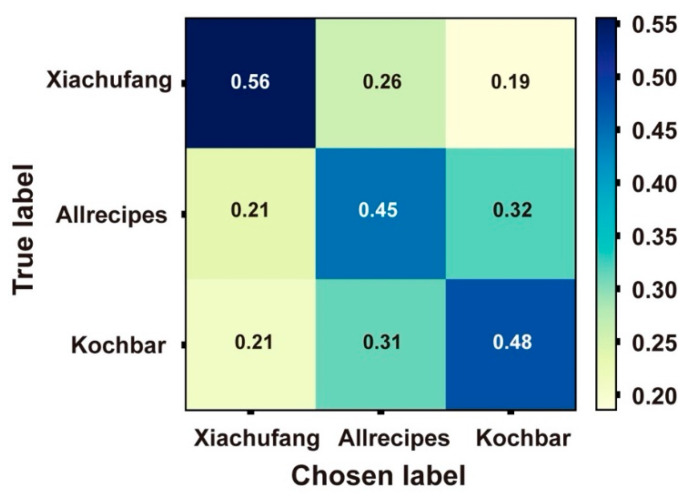
Confusion matrix of participants’ judgements.

**Figure 6 foods-09-00823-f006:**
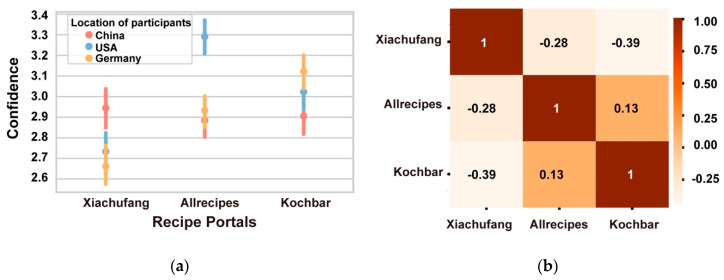
Participant confidence in labels across recipe portals. (**a**) Mean value and error bar for confidence ratings for each collection by participants from different locations. (**b**) Correlation matrix for participant confidence scores for their labels for different recipe portals.

**Figure 7 foods-09-00823-f007:**
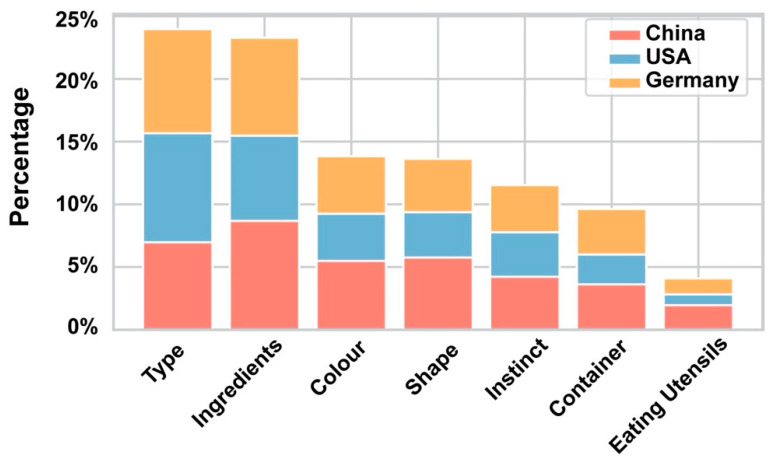
The percentage based on frequency of each single factor chosen by the participants.

**Figure 8 foods-09-00823-f008:**
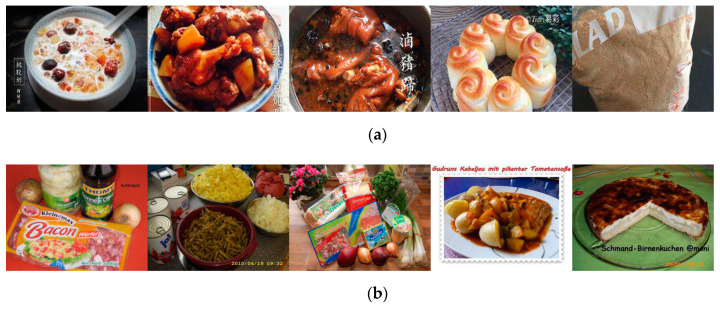
Examples of images with text. (**a**) Images with Chinese characters from *Xiachufang.com*. (**b**) Images with German characters from *Kochbar.de*.

**Figure 9 foods-09-00823-f009:**
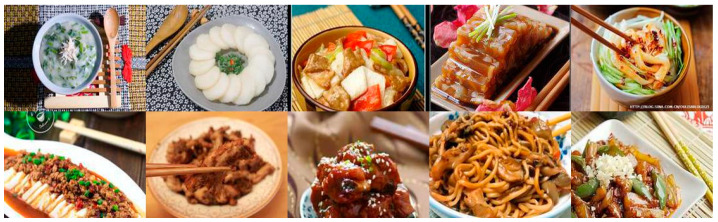
Examples of images with eating utensils from *Xiachufang.com*.

**Table 1 foods-09-00823-t001:** Survey questions for the participants.

Question	Scale
Personal information	
Age	<18, 18–24, 25–34, 35–44, 45–55, >55
Gender	Male, Female, Other
Nationality	Select from a drop-down list
Experiences with the recipe portals	
Familiarity with each recipe portal	Likert scale 1 (Not at all)–5 (Very familiar)
Frequency of using recipe portals	Hardly use, At least once every three months, At least once per month, At least once per week, Use on a daily basis
Settlement and travel experience	
Experience in China	Never visited, I have been there once or a few times, I visit or have visited regularly, I have lived there for many months or longer, I am a permanent resident
Experience in USA	Never visited, I have been there once or a few times, I visit or have visited regularly, I have lived there for many months or longer, I am a permanent resident
Experience in Germany	Never visited, I have been there once or a few times, I visit or have visited regularly, I have lived there for many months or longer, I am a permanent resident
Frequency of cross-continental travelling	Never, Less than once per year, 1–2 times per year, More than 2 times per year
Interests in food/recipes from foreign cultures	
Interest in food/recipes from other cultures	Likert scale 1 (No interest at all)–5 (Very interested)
Frequency of trying food/recipe from other cultures	Hardly ever, Less than once per month, At least once per month, At least once per week, Most days
Free-text field	Blank space left for all participants

**Table 2 foods-09-00823-t002:** Prediction accuracy for recipe source different visually related feature sets. Best performing scores for each classifier are bolded. NB = naive Bayes; LOG = logistic regression; RF = random forest; EVF = explicit visual features; LBP = local binary patterns; SIFT = scale-invariant feature transform; DNN = deep neural network.

Features	Accuracy		
	NB	LOG	RF
EVF (Brightness)	0.41	0.41	0.42
EVF (Sharpness)	0.41	0.41	0.43
EVF (Contrast)	0.37	0.37	0.42
EVF (Colourfulness)	0.38	0.38	0.41
EVF (Entropy)	0.38	0.37	0.40
EVF (RGB contrast)	0.38	0.38	0.41
EVF (Sharpness variation)	0.41	0.41	0.41
EVF (Saturation)	0.39	0.39	0.40
EVF (Saturation variation)	0.39	0.38	0.41
EVF (Naturalness)	0.38	0.38	0.40
EVF (All features)	0.47	0.54	0.55
Colour histogram	0.43	0.52	0.54
LBP	0.48	0.52	0.52
SIFT	0.58	0.72	0.67
DNN	0.67	0.86	0.78
All features	0.73	0.89	0.85

**Table 3 foods-09-00823-t003:** Results when predicting recipe image source and participant-applied label based on different visual properties and other factors. Best performing scores for each classifier are bolded. NB = Naive Bayes; LOG = Logistic Regression; RF = Random Forest.

	Accuracy					
	NB		LOG		RF	
	Recipe’s Origin	Participants’ Judgements	Recipe’s Origin	Participants’ Judgements	Recipe’s Origin	Participants’ Judgements
EVF (Brightness)	0.43	0.36	0.41	0.33	0.41	0.34
EVF (Sharpness)	0.41	0.36	0.43	0.37	0.44	0.36
EVF (Contrast)	0.37	0.34	0.37	0.34	0.35	0.34
EVF (Colourfulness)	0.41	0.34	0.40	0.34	0.40	0.34
EVF (Entropy)	0.38	0.36	0.38	0.36	0.39	0.36
EVF (RGB Contrast)	0.37	0.34	0.38	0.35	0.37	0.35
EVF (Sharpness variation)	0.42	0.36	0.43	0.36	0.42	0.37
EVF (Saturation)	0.42	0.32	0.42	0.34	0.41	0.34
EVF (Saturation variation)	0.39	0.36	0.39	0.34	0.39	0.37
EVF (Naturalness)	0.39	0.36	0.40	0.36	0.40	0.34
EVF (All features)	0.50	0.38	0.56	0.38	0.55	0.38
Colour histogram	0.37	0.34	0.49	0.36	0.54	0.38
LBP	0.47	0.38	0.50	0.38	0.51	0.39
SIFT	0.57	0.40	0.52	0.39	0.65	0.44
DNN	0.66	0.43	0.82	0.42	0.77	0.45
All features (Visually)	0.69	0.43	0.85	0.43	0.84	0.46
Ingredients	0.34	0.35	0.34	0.35	0.34	0.35
Type	0.34	0.35	0.34	0.35	0.34	0.35
Colour	0.35	0.34	0.35	0.34	0.35	0.34
Shape	0.33	0.33	0.32	0.33	0.32	0.33
Container	0.34	0.36	0.34	0.36	0.34	0.36
Eating utensils	0.35	0.36	0.35	0.36	0.35	0.36
Instinct	0.35	0.36	0.35	0.36	0.35	0.36
All factors	0.34	0.38	0.35	0.37	0.35	0.36

**Table 4 foods-09-00823-t004:** Top ten factors or combinations of factors indicated by participants to have influenced the label applied.

Factors	Count	Percentage
Ingredients, Type	226	84%
Type	226	84%
Ingredients	164	61%
Instinct	127	47%
Ingredients, Colour, Type	94	35%
Shape, Type	76	28%
Ingredients, Shape, Type	76	28%
Ingredients, Type, Instinct	75	28%
Ingredients, Colour	62	23%
Type, Instinct	62	23%

**Table 5 foods-09-00823-t005:** Coding scheme for factors reported by participants.

Categories		N ^1^	Description	Examples ^2^
Food Factors	Adjective	24	Participants left single adjective to describe the food in the recipe image	GE_96 ^3^: good US_98: healthy
	Style	26	Participants reported how the food looked in the recipe image	CH_30: Chinese dish is generally not so ugly US_85: Plate design GE_1: Size of the food
	Ingredients	17	Participants reported at least one ingredient they saw in the recipe image	CH_10: There is rice US_95: The egg on top looks like oriental food. GE_58: Contains coriander and Chili?
	Cooking methods	5	Participants reported how to cook the food in the recipe image	CH_13: Production methods, it’s barbecue
Non-food factors	Text	49	Participants reported the letters, characters or water marks, etc. they saw in the recipe image	CH_42: “猪肉” is Chinese character US_77: German writing GE_64: Date format: 19.02.2013 is German
	Object/Background	16	Participants described the objects or setting in the recipe image instead of the food itself	CH_30: Stairs US_55: Newspaper GE_31: Kitchen utensils
	Photo	9	Participants described the photographic and post-processing of the recipe image instead of the food itself	CH_51: A popular filter was used US_72: Angle of the photo, light in the photo GE_39: Bad lighting
	Personal experience	2	Participants reported their own experience with the food in the recipe image	US_5: I know this type of food CH_41: It seems like I’ve eaten this
	Unknown	18	Participants left comments but offered deficient information	CH_41: It could come from any portal US_3: not sure what type of food that is GE_96: nothing

^1^ Column N indicates how many times each kind of factor was reported by the participants; ^2^ Column Examples indicates the ID of participants and the comments they left; ^3^ Participant’s ID comprising their location (CH: China, US: the US, GE: Germany) and a number.

**Table 6 foods-09-00823-t006:** Logistic regression model of participants’ judgements.

	Dependent Variable Correct/Wrong Answer		
	Coef(*β*)	95% CI	OR
Constant	−0.192	[−0.364, −0.020]	0.825
Ingredients	0.069	[−0.085, 0.223]	1.071
Type	0.184 *	[0.031, 0.338]	1.202 *
Colour	0.031	[−0.134, 0.196]	1.031
Shape	−0.063	[−0.229, 0.102]	0.939
Container	0.013	[−0.170, 0.196]	1.013
Eating utensils	0.394 **	[0.132, 0.657]	1.483 **
Instinct	0.008	[−0.163, 0.178]	1.008
McFadden’s R^2^	0.004		
Log likelihood	−1863.5		
AIC	3743		

Note: Coef, coefficient; *, *p* < 0.05; **, *p* < 0.01

**Table 7 foods-09-00823-t007:** Ordinal regression models predicting participant confidence for images associated with each recipe portal.

	Dependent Variable								
	Confidence on *Xiachufang*			Confidence on *Allrecipes*			Confidence on *Kochbar*		
	Coef(*β*)	95% CI	OR	Coef(*β*)	95% CI	OR	Coef(*β*)	95% CI	**OR**
Ingredients	0.009	[−0.126, 0.145]	1.009	−0.098	[−0.233, 0.038]	0.907	−0.220 **	[−0.356, −0.839]	0.803 **
Type	−0.294 ***	[−0.430, −0.158]	0.745 ***	−0.030	[−0.167, 0.105]	0.970	−0.031	[−0.167, 0.104]	0.970
Colour	0.156 *	[0.009, 0.302]	1.168 *	−0.147 *	[−0.294, −0.000]	0.863 *	−0.102	[−0.249, 0.044]	0.903
Shape	0.010	[−0.137, 0.156]	1.010	−0.145	[−0.292, 0.001]	0.865	−0.004	[−0.151, 0.142]	0.996
Container	0.241 **	[0.078, 0.405]	1.273 **	−0.011	[−0.172, 0.151]	0.990	−0.143	[−0.306, 0.020]	0.867
Eating utensils	0.365 **	[0.123, 0.608]	1.440 **	−0.258 *	[−0.489, −0.027]	0.772 *	−0.177	[−0.413, 0.060]	0.838
Instinct	−0.208 **	[−0.360, −0.057]	0.812 **	−0.198 *	[−0.349, −0.047]	0.820 *	−0.093	[−0.245, 0.060]	0.912
MacFadden’s R^2^	0.006			0.003			0.002		
Log likelihood	−4256.70			−4248.05			−4233.68		
AIC	8535.41			8518.09			8489.36		

Note: Coef, coefficient; *, *p* < 0.05; **, *p* < 0.01; ***, *p* < 0.001.

**Table 8 foods-09-00823-t008:** Comparison of classification accuracy achieved by different groups based on demographic information. Only attributes with significant results are included in the table. Statistical significance across groups was determined using Mann–Whitney U tests.

	Overall Accuracy	Accuracy on *Xiachufang*	Accuracy on *Allrecipes*	Accuracy on *Kochbar*
	Mean (+/− Std)	Mean (+/− Std)	Mean (+/− Std)	Mean (+/−Std)
Gender				
Male	0.49(+/−0.17)	0.51(+/−0.29)	0.44(+/−0.28)	0.51(+/−0.30) *
Female	0.50(+/−0.18)	0.61(+/−0.28) **	0.46(+/−0.28)	0.44(+/−0.31)
Age				
Age <35	0.50(+/−0.18)	0.59(+/−0.29) **	0.41(+/−0.27)	0.50(+/−0.30) *
Age ≥35	0.48(+/−0.17)	0.49(+/−0.29)	0.52(+/−0.27) ***	0.50(+/−0.30)
Experience of each country (China)				
Never visited–been there a few times	0.49(+/−0.17)	0.51(+/−0.29)	0.47(+/−0.27) *	0.49(+/−0.29)
Visit regularly–permanent resident	0.50(+/−0.18)	0.63(+/−0.28) ***	0.41(+/−0.29)	0.45(+/−0.31)
Experience of each country (The US)				
Never visited–been there a few times	0.49(+/−0.18)	0.61(+/−0.29) ***	0.39(+/−0.28)	0.49(+/−0.31)
Visit regularly–permanent resident	0.48(+/−0.17)	0.47(+/−0.27)	0.53(+/−0.26) ***	0.46(+/−0.30)
Experience of each country (Germany)				
Never visited–been there a few times	0.48(+/−0.18)	0.56(+/−0.27)	0.46(+/−0.28)	0.43(+/−0.31)
Visit regularly–permanent resident	0.50(+/−0.17)	0.55(+/−0.31)	0.43(+/−0.28)	0.54(+/−0.29) ***
Familiarity with each recipe portal (*Xiachufang.com*)				
Not familiar (≥2 on Likert scale) Familiar (≤3 on the Likert scale)	0.51(+/−0.17) ** 0.46(+/−0.17)	0.55(+/−0.29) 0.57(+/−0.31)	0.46(+/−0.28) 0.42(+/−0.28)	0.52(+/−0.29) *** 0.39(+/−0.31)
Familiarity with each recipe portal (*Allrecipes.com*)				
Not familiar (≥2 on Likert scale) Familiar (≤3 on the Likert scale)	0.50(+/−0.17) 0.48(+/−0.17)	0.62(+/−0.28) *** 0.48(+/−0.28)	0.40(+/−0.28) 0.50(+/−0.27) ***	0.50(+/−0.29) 0.46(+/−0.31)
Familiarity with each recipe portal (*Kochbar.de*)				
Not familiar (≥2 on Likert scale) Familiar (≤3 on the Likert scale)	0.50(+/−0.17) 0.48(+/−0.18)	0.58(+/−0.28) * 0.50(+/−0.32)	0.44(+/−00.28) 0.46(+/−0.28)	0.48(+/−0.30) 0.48(+/−0.31)
Interest in food from foreign cultures				
Not interested (≥2 on Likert scale)	0.41(+/−0.23)	0.46(+/−0.28)	0.33(+/−0.33)	0.45(+/−0.39)
Interested (≤3 on the Likert scale)	0.50(+/−0.17) *	0.56(+/−0.29) *	0.46(+/−0.27) *	0.48(+/−0.30)
Interest in recipes from foreign cultures				
Not interested (≥2 on Likert scale) Interested (≤3 on the Likert scale)	0.45(+/−0.23) 0.50(+/−0.17) *	0.50(+/−0.27) 0.56(+/−0.29)	0.37(+/−0.33) 0.46(+/−0.27) *	0.47(+/−0.34) 0.48(+/−0.30)
Frequency of trying recipes from other cultures				
Once per month	0.48(+/−0.18)	0.58(+/−0.29) *	0.41(+/−0.28)	0.46(+/−0.29)
Once per month	0.50(+/−0.17)	0.52(+/−0.29)	0.49(+/−0.27) **	0.50(+/−0.32) **

Note: Std, standard deviation, *, *p* < 0.05; **, *p* < 0.01; ***, *p* < 0.001.

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
