# Peer review of "Visual Cultural Biases in Food Classification"

_foods, 2020, doi:10.3390/foods9060823_

Round 1

Reviewer 1 Report

I would like to thank the authors because they've addressed all my concerns. I would only suggest to edit Table 7 because it seems not well formatted.

Author Response

Response to Review 1 Comments

Point 1: I would only suggest to edit Table 7 because it seems not well formatted.

Response 1: Thank you, we have done this. We think that the submission process
changed the presentation of the tables somehow.

Reviewer 2 Report

I'd like to thank the authors for revising the manuscript.

Some minor comments below

Line 61, consider separating this as a new paragraph after the bullet points.

Line 200 - Data analysis, this section is still missing parts of analysis that was carried out by the authors. Take confusion matrix for example, please revise. This needs to be revised since lots of analysis that the authors had carried out isn't explained well here.

Some comments were not addressed from the previous review.

Instead of having RQ1-3 as a header, say what was the important highlight for this specific research question. Add relevant references and suggest mechanisms to support the findings.

More information is also on how the authors arrive on the type of questions that they have on the questionnaire. For example, on Q3 why did the author prioritise ingredients, colour, and shape of food?

Author Response

Response to Review 2 Comments

Point 1: Line 61, consider separating this as a new paragraph after the bullet points.

Response 1: Thank you, we have modified this.

Point 2: Line 200 - Data analysis, this section is still missing parts of analysis that was carried out by the authors. Take confusion matrix for example, please revise. This needs to be revised since lots of analysis that the authors had carried out isn't explained well here.

Response 2: We have added text to explain confusion matrices and why these have
been used. We also added text explaining the metric used to measure both human and algorithm performance. All analyses performed are therefore now described in this paragraph.

Point 3: Instead of having RQ1-3 as a header, say what was the important highlight for this specific research question. Add relevant references and suggest mechanisms to support the
findings.

Response 3: We have removed the headers (RQ1-3) and have integrated the
relationship between the discussion and the specific research questions into the text.

We have also added text relating the findings to the literature. We hope this satisfies the reviewer’s comment.

Point 4: More information is also on how the authors arrive on the type of questions that they have on the questionnaire. For example, on Q3 why did the author prioritise ingredients, colour, and shape of food?

Response 4: We have added more descriptions of the factors we listed in RQ3 andexplained why we chose them. The reasons were typically based on the literature or the results of our pilot study. We hope the added text makes this clear.

This manuscript is a resubmission of an earlier submission. The following is a list of the peer review reports and author responses from that submission.

Round 1

Reviewer 1 Report

Introduction

Line 35 - Consider incorporating the word objective into the sentence.

Line 45 - The wording ground truth isn't really needed here.

Line 50 - Is this missing a bullet point?

Section 1.1

Incorporate together the Introduction section, there is no need to separate this section alone.

Section 1.2 - please delete as it is not required b the journal I believe.

Section 2.1 - provide the company who owns these sites.

Line 91 - Add what site for OpenIMAJ and what company

Section 2.3.1

Where did the author host the online survey?

Line 186 - Why is there a \ before %.

Line 190 - Be explicit what social media was used in DE.

Line 193 - Add company who owns the platform.

Figure 2. The US = Change accordingly to USA

The authors had also failed to mention on why and how the selections of the survey was generated. Was it through a pilot study? Add more details.

Under stats analysis - it seems that the authors had omitted quite a lot of details. For example, numerous regression techniques was used here (see Table 1), but wasn't mentioned. Add more details.

Free-text analysis method should also be included here.

Figure 2. Elaborate why confusion matrix was considered in this paper.

Make sure for R2 - the 2 is superscripted.

Line 379 - delete \textit

Table 7. What stats method did the author use to suggest that there are difference? Add more details.

Discussion

Instead of saying RQ1-3, say what was the important highlight for this specific research question. Add relevant references and suggest mechanisms to support the findings.

Section 4.5

Considering the limitations of the study - what will be the recommended next step in this research area?

Reviewer 2 Report

The paper Visual Cultural Biases in Food Classification deals with an interesting topic and it has some peculiarities. Overall, the paper is well written and results are well presented. I only have one major concern and some minors that I will explain with bullet points: 

MAJOR CONCERN

Introduction section must be improved by citing more relevant literature. For example, in lines 32-33 authors write “The reasons why this is the case, however, are not particularly well understood”; from my point of view, a recap about this point is mandatory. What does other authors say about this topic? Any relevant finding? Furthermore, I also do not like the fact that three subparagraphs are presented within the introduction section. For instance, lines 47-48-49 are basically saying the same of lines 53, 55, 57. Please amend.Lastly, I think that the beginning of line 35 contains a typo (“Objective” has to be in italic since it was a subparagraph, am I right?)

MINOR CONCERNS

  • Paragraph number 2 is very well written and understandable, also from a non-expert in the field. I only have some suggestions:
    • Figure 1 is very small and almost unreadable. I suggest putting a Word file in appendix, not an image;
    • Lines 165-166 should be in a past tense as line 167 is;
    • Lines 168-179. I think that these info could be better understandable in a table. Please amend;
    • Line 186. There is a typo after the 98;
    • Figure 2. I think that a table could be just fine. I really don’t like figures made with Excel. If you decide to keep it, could you try using a different software? The image is not sharp, it’s a bit blurry;
  • Line 220 has a typo. A double “minus” between 0.47 and 0.55;
  • Line 233. There is a problem with the comas used. The phrase should be “In summary, the experiments show that… “
  • Figure 4. Again, a bit blurry and the (b) is cut on the right side. The two images should be aligned too.
  • Lines 275 – 282 are comments on the results and should be in the discussion section.
  • Lines 307 contains an editing error.
  • Lines 386-388 contains some English editing problems. Please amend
  • Table 6. The CI makes the table a bit difficult to read and creates confusion with the title of columns (which is also Confidence on…). In addition, there are some editing problems. A revision of this table is needed.
  • Table 7 also has some editing problems.
  • The conclusion section is unnecessary. I suggest to join paragraph 4 and 5 in one.

Therefore, I suggest major revisions in order to improve the overall merit of the work and I will be glad to read the revised version.